# Red Blood Cell Donor Sex Associated Effects on Morbidity and Mortality in the Extremely Preterm Newborn

**DOI:** 10.3390/children9121980

**Published:** 2022-12-16

**Authors:** Tara M. Crawford, Chad C. Andersen, Michael J. Stark

**Affiliations:** 1The Robinson Research Institute, The University of Adelaide, Adelaide 5006, Australia; 2The Department of Neonatal Medicine, The Women’s and Children’s Hospital, North Adelaide 5006, Australia

**Keywords:** red blood cell transfusion, blood donor sex, outcomes, preterm newborn

## Abstract

Transfusion exposure increases the risk of death in critically ill patients of all ages. This was thought to relate to co-morbidities in the transfusion recipient. However, donor characteristics are increasingly recognised as critical to transfusion recipient outcome with systematic reviews suggesting blood donor sex influences transfusion recipient health. Originally focusing on plasma and platelet transfusions, retrospective studies report greater risks of adverse outcomes such as transfusion related acute lung injury in those receiving products from female donors. There is increasing awareness that exposure to red blood cells (RBCs) poses a similar risk. Recent studies focusing on transfusion related outcomes in extremely preterm newborns report conflicting data on the association between blood donor sex and outcomes. Despite a renewed focus on lower versus higher transfusion thresholds in neonatal clinical practice, this group remain a heavily transfused population, receiving on average 3–5 RBC transfusions during their primary hospital admission. Therefore, evidence supporting a role for better donor selection could have a significant impact on clinical outcomes in this high-risk population. Here, we review the emerging evidence for an association between blood donor sex and clinical outcomes in extremely preterm newborns receiving one or more transfusions.

## 1. Introduction

The risk of death in critically ill patients, both short- and long-term mortality increases with each red blood cell (RBC) transfusion exposure [1,2]. In the extremely preterm newborn, one of the most heavily transfused patient groups [3], observational data has proposed an association between RBC transfusion and adverse neonatal outcomes, although the data remains limited [4,5]. Morbidities associated with RBC transfusion exposure, include necrotizing enterocolitis (NEC), bronchopulmonary dysplasia (BPD), retinopathy of prematurity (ROP), and intraventricular haemorrhage (IVH) [6]). However, limitations to much of the primary evidence has resulted in ongoing controversy about the contribution transfusion exposure has to the pathophysiology of each morbidity [7].

The preterm newborn has a developmentally unique immune system which contributes to their vulnerability to adverse transfusion-related effects. Preterm birth is associated with multifaceted abnormalities, resulting in multi-system immaturity. Chronic fetal inflammation, frequently without a definite cause, is associated with preterm birth and preterm neonate complications [8], with fetal inflammation the greatest risk factor for subsequent neonatal morbidity [9]. Proteomic analysis has reported changes in the balance between inflammation and immunomodulation status throughout the neonatal period [10]. As a result, the preterm newborn is particularly susceptible to inflammatory injury, with resultant increases in pro-inflammatory cytokines in peripheral blood plasma [11,12]. This inflammatory state is associated with greater disposition to a range of inflammatory pathologies in neonatal tissues, most notably the lungs, gastrointestinal tissues, and brain [13,14,15]. Transfusion exposure compounds this by directly modulating immune cell function, a process referred to as transfusion related immunomodulation (TRIM). TRIM has been proposed as a “two-insult” hypothesis, with repeat transfusion exposure compounding long-term or permanent alteration in immune function [16]. The initial transfusion exposure sensitizes normal inflammatory processes by priming host neutrophils, each subsequent exposure results in an amplified immune response in the recipient [17]. The exact mechanism/s are yet to be fully characterized, however is it likely biological response mediators that accumulate during RBC storage and the subsequent interaction with the recipients immune function. The high oxygen consumption, weak antioxidant system, and the inability to induce antioxidant defences results in the preterm newborn being particularly susceptible to oxidative stress and damage [18]. Oxidative stress is magnified by other predisposing conditions, such as hyperoxia, hypoxia, ischemia, hypoxia–reperfusion, and, critically, inflammation. Furthermore, in line with a number of neonatal intensive care treatments, RBC transfusion enhances free radical production and oxidative stress [19], and therefore end organ oxidative damage [20].

Research efforts have predominantly focused on factors influencing blood quality to improve post-transfusion clinical outcomes. Much attention has been given to the age of transfused blood and the development of storage lesions [21]. Storage lesion describes the gradual deterioration of various components of the RBCs following their isolation and extended hypothermic storage, resulting in oxidative stress and metabolic impairments of the red cell. Despite the potential for an etiological link between storage lesion and adverse clinical sequelae in the transfusion recipient [22], the use of RBCs less than 7 days old compared with standard blood bank practice (up to 42 days old) did not improve outcomes in premature, very low-birth-weight infants requiring a transfusion [23]. However, pre-storage leukodepletion, designed to limit the biological effects leukocytes present in stored blood products may have, has proven more successful with improvements in several clinical outcomes in preterm newborns requiring RBC transfusions [24].

More recently, interest has turned to the influence of donor characteristics on adverse post-transfusion responses [25]. Initially focusing on transfusion-related acute lung injury (TRALI) following plasma and platelet transfusion from a female donor, recent observational studies have also highlighted similarly increased mortality in transfusion recipients of RBCs from female donors [26,27,28]. While the effects of donor characteristics, including donor sex, on transfusion outcomes has been systematically reviewed [29], this meta-analysis of all patient groups included only two studies in newborns [30,31]. 

Here we review recent additions to the neonatal literature focusing on the role of donor sex on transfusion recipient outcomes. Furthermore, we discuss the underlying biological mechanisms that could potentially contribute to the association between donor sex and incidence of significant neonatal morbidities and mortality. Finally, we address design considerations needed for future large-scale studies of the impact of transfusion exposure in this highly transfused population. Such studies are required to definitively address uncertainty about the safety of transfusion in the preterm newborn and will represent important steps towards personalised transfusion therapy. 

## 2. Defining Donor Sex Exposure

Two types of donor sex comparison are commonly reported in the adult literature: single donor sex exposure and donor-recipient sex mismatch. For instance, transfusion with RBCs from female donors has been reported to be associated with greater mortality rates [26,32,33], an association also seen with donor-recipient sex-mismatched transfusion with female donor RBCs [27,34]. However, this inconsistency in the donor sex comparisons investigated and the inclusion of patients who have received RBCs from both male and female donors, has limited interpretation of the effects attributable to a specific donor sex. Studying only patients who received one transfusion would simplify patient exposure classification as ‘sex-matched’ or ‘sex-mismatched’ while avoiding the confounding effect of number or transfusions received. This, however, results in inclusion of only less unwell patients in any analysis.

Criticism of the data generated from these studies has focused on residual confounders, resulting in model misspecification [26], large proportions of missing data [33], and the introduction of selection bias through informative censoring [35]. Other possible explanations for inconsistency in the findings could be related to the fact that some include all types of blood transfusion exposures, including plasma, platelets, and RBCs [32,36], whereas others focus solely on RBC transfusions [26,27,33,37]. Finally, most of the adult literature uses mortality as the transfusion outcome, with less attention paid to the possibility of donor-related factors being associated with morbidity outcomes. 

## 3. Evidence in the Preterm Newborn for Donor Sex Effects on Transfusion Recipient Outcomes

The first report of donor sex specific outcomes in the preterm newborn was by Paul and colleagues, who described a retrospective cohort of 2311 patients in whom 122 (5.3%) developed NEC [31]. Of these, 33 (27%) of cases occurred within 48 h of an RBC transfusion. In this group, 84% of RBC units transfused were from male donors compared with 62% in neonates that developed necrotizing enterocolitis greater than 48 h after transfusion (*p* = 0.03). However, the proportion of sex matched versus mismatched transfusions in each group was not reported. While this study focuses on an outcome other than mortality, the association between RBC transfusion exposure and NEC remains controversial. The most recent meta-analysis of available observational data found no such effect [38], while a systematic review of randomized trials on blood transfusion thresholds in extremely low birth weight infants also fails to support an association between transfusion and NEC [39]. 

More recently, Murphy and colleagues reported the findings of an exploratory, retrospective, cohort study which aimed to investigate the potential of RBC donor sex on a range of neonatal outcomes including BPD, IVH grade III or IV, spontaneous intestinal perforation (SIP), NEC Bell criteria stage 2 or greater, and death [40]. The study population, comprising 170 patients < 32 weeks’ gestation, was divided to allow for a comparison of recipients of either sex who received either male and/or female blood with recipients of either sex who received male blood only. While transfusion with any female blood was significantly associated with any morbidity (OR 2.35 95% CI 1.11–4.95, *p* = 0.025) and a longer length of stay (20.6 days in NICU, *p* = 0.01), this effect was no longer present following adjustment for the number of transfusions received. The potential for a dose-response effect of female blood was also investigated with an increase in the odds of any morbidity for each additional transfusion of female RBCs observed (OR 2.63 95% CI 1.21–5.7, *p* = 0.015). 

While this study has several strengths, including the exclusion of patients who received other blood products and assessment of the Morbidity Assessment Index for Newborns score on days 1 and 7 to limit the potential for confounding by illness severity, clear limitations remain. As for all studies of this design, no consideration of transfusion timing and its relation to diagnosis of the outcome/s of interest was possible. Clearly, a transfusion cannot be causative if it is administered as part of the therapeutic response to a clinical event. In addition, extrapolation of transfusion exposure to later development of BPD, defined at 36 weeks by conventional criteria or the need for pressure support >2 L/min [40], is problematic. While TRALI is accepted as the leading transfusion-related complication, prior to the Montreaux definition of this condition (based on timing, lung imaging, absence of congenital heart disease including the patent ductus arteriosus, and an oxygen deficit measured by an oxygenation index) [41], newborn infants in the first month of life were excluded. While it is plausible that the disruption to the inflammatory signaling pathway from transfusion exposure in the at-risk preterm infant may predispose or magnify the development of BPD, the contributory effect of transfusion exposure has not been recognized [42], this represents a considerable amplification of the original definition of TRALI. Interestingly, the “female donor” group received a significantly greater number of transfusions. While the mechanistic basis of the association between donor sex and adverse clinical outcomes were not a focus of this study, it is now recognized that an RBC transfusion is biologically active, resulting in immune activation and inflammation in the recipient [43,44], effects magnified by repeat transfusion exposure [17].

A different approach was taken by Patel and colleagues [45]. Here the authors included newborns previously recruited into the prospective Transfusion-Transmission of Cytomegalovirus cohort study [46]. Eligible subjects were very low birth weight (<1500 g) newborns who received an RBC transfusion from only female donors or exclusively male donors. Those who received transfusions from both sexes were excluded. A total of 181 newborns, received 499 transfusions and included in the analysis. Female donors represented 136 (27%) of the transfusions. The exclusively female donor group had a lower risk of death or serious morbidity (21% vs. 45%, adjusted OR 0.26 95% CI 0.09–0.65) [45]. Further, in a subset of 76 newborns a single RBC transfusion from a female donor was associated with the lowest risk of the primary composite outcome (RR 0.15, 95% CI 0.02–0.99) after adjustment for birthweight and donor age. Intriguingly, the authors report a reduction in the protective effect of exposure to female donor RBCs with an increasing number of transfusion exposures. This could be interpreted as simply those newborns receiving the most transfusions being the most likely to die or survive with morbidity [47]. However, it could also reflect a cumulative effect of transfusion exposure on transfusion recipient responses that contribute to the pathophysiology of neonatal morbidities [17]. 

Why are the findings of these studies’ contradictory while many of the limitations are common to both? The most obvious methodological difference is the inclusion, rather than exclusion, of infants who received both male and female donor RBCs in the Murphy study [40]. Patel and colleagues argue that newborns who receive transfusions from both donor sexes, suggesting greater transfusion exposure, may have greater illness severity [45]. When comparing transfusion exposure between those newborns included versus those excluded due to exposure to both male and female donor RBC in their cohort, the difference in the median number of transfusions was of a factor of 3. However, both studies found an adverse interaction between female donor sex and the number of transfusions, resulting in an increase in the odds of any morbidity [40] versus a reduction in the protective effect of exposure to female donor RBCs for each additional transfusion [45]. 

Finally, the primary outcome differs between the two studies, potentially affecting any association between donor sex and morbidity and mortality. While IVH was included in the composite “any morbidity” outcome in the study by Murphy and colleagues, it was excluded in the Patel study. This was justified by the authors on the grounds that all transfusions were evaluated, including those occurring after the typical timing of IVH occurrence in their population of very low birth weight newborns [45]. 

## 4. How Might Donor Sex Influence Outcome?

Currently, the biological mechanisms which underlie the association of donor sex and increased risk of adverse outcomes following exposure to RBCs remain poorly characterized. What is likely, however, is that multiple factors contribute to the influence of donor sex on recipient outcomes [27].

### 4.1. Inherent Differences in RBCs from Male and Female Donors

RBC transfusion packs from male and female donors are inherently different with haematological and redox parameters differing between male and female donors during storage. Female donors have a greater proportion of younger RBCs at the time of donation, a difference greatest during reproductive age, when more young RBCs enter the circulation to compensate for blood loss through menstruation [48]. As a result, cell deformability and oxygen carrying capacity are greatest in RBCs from female donors [49,50] however, RBC mechanical fragility is lower [51]. Conversely, RBC transfusion packs from male donors have higher hemoglobin content, resulting in larger increments in hemoglobin concentration [52], but undergo higher rates of storage hemolysis [49,52]. There are also sex-specific differences in RBC membrane properties during storage [53]. RBC membrane expression of CD47 falls significantly with time in RBCs from male donors while membrane lipid leakage is greater, suggesting conformational changes in the RBC membrane and phospholipid bilayer destabilization leading to greater RBC clearance [54]. Finally, female donor RBC units have greater preservation of intracellular antioxidant capacity, lower levels of intracellular reactive oxygen species, and decreased levels of oxidative hemolysis compared to those from male donors [55,56].

There is some evidence that these differences may result in sex specific responses in the transfusion recipient. For female transfusion recipients, the delivery of larger amounts of free haemoglobin with RBC transfusions from male donors may result in the haptoglobin scavenging capacity of the reticuloendothelial system being overwhelmed [57]. Excess free haemoglobin is also scavenged by endothelial derived nitric oxide, producing methaemoglobin, a reduction in nitric oxide bioavailability and endothelial dysfunction and oxidative injury [57,58]. This increase in methaemoglobin levels has also been shown to decrease tissue oxygen delivery with resultant at end organ hypoxia [59].

### 4.2. Transfusion Related Immunomodulation and the Influence of Donor Sex

We and others have reported an increase in circulating plasma pro-inflammatory cytokines and markers of endothelial activation concentration between 2 and 48 h after single transfusion exposures in the weeks following preterm birth [43,44]. Furthermore, transfusion related pro-inflammatory immune responses increase in magnitude with repeat transfusion exposure in preterm newborns [17]. These responses may represent a biologically plausible pathway whereby RBC transfusion promotes immune cell activation and pro-inflammatory responses [44], processes that may result in adverse consequences for the transfusion recipient. 

How donor sex influences these immune responses is an area of growing interest, with immune cell activation by HLA antibodies [60], which are critical to induction and regulation of immune responses, identified as one potential candidate [61]. HLA antibodies are more common in women, with sensitization rates increasing significantly with a history of pregnancy of 20–50% [62]. Evidence linking donor derived HLA antibodies and donor sex to clinical outcomes centres on TRALI, where female donor plasma is a known risk factor [63]. In TRALI, HLA antibodies activate neutrophils in the pulmonary vasculature, releasing pro-inflammatory cytokines, including IFNγ, IL-6 and IL-8 [64,65], resulting in end-organ damage [66]. Whether these mechanisms are influenced by donor sex in neonatal transfusion recipients is currently unknown.

Intriguingly, there is emerging evidence that preterm newborns may themselves exhibit sex-specific inflammatory responses to RBC transfusion [67]. In an analysis of 19 cytokines and inflammatory biomarkers measured pre- and post-transfusion in preterm newborns enrolled in the Transfusion of Prematures (TOP) trial [68], Benavides and colleagues reported significantly greater increases with each additional transfusion in monocyte chemoattractant protein (MCP-1) in females but not in males [67]. In addition, higher concentrations of MCP-1 were associated with worse neurodevelopmental outcomes determined by Bayley-III assessment. Future studies in considerably larger cohorts of newborns are clearly necessary to delineate any direct associations of inflammatory biomarkers and adverse outcomes. However, this data highlights the hurdles in understanding the complex interactions between donor sex, RBC transfusion and preterm newborn immunomodulatory responses. 

### 4.3. Transfusion Related Endothelial Activation and the Influence of Donor Sex

Donor sex related immunomodulation, with increases in pro-inflammatory cytokines, also contributes to endothelial activation. This is a potential predictive marker for transfusion associated adverse outcomes and has recently been shown to be increased with sex mismatched RBC transfusion exposure [69]. In vitro data supports the release of markers of endothelial activation following the incubation of endothelial cells with blood product supernatant [70,71]. Clinical data reports increased ICAM-1, a marker of endothelial activation, and syndecan-1, which is released with the loss of endothelial glycocalyx integrity, following RBC transfusion [72]. The link between transfusion exposure and endothelial activation may be directly related to post-transfusion increases in pro-inflammatory cytokines [73], a response observed in the preterm newborn [43,44], or due to the presence of free heme and non-transferrin bound iron [74,75]. Recently, Alshalani and colleagues reported significantly greater post-transfusion concentrations of syndecan-1 and soluble thrombomodulin in adult intensive care patients who received a single sex-mismatched RBC transfusion compared to those who received sex-matched blood [69]. Critically, both syndecan-1 and soluble thrombomodulin are known predictors of in-hospital mortality [76]. Again, this clinical data has been generated from adult studies, with evidence in preterm newborns lacking. 

Extracellular vesicles, specifically microparticles, transfused along with the RBCs are biological response modifiers that may play a significant immunomodulatory role in the transfusion recipient [77,78]. They have been shown to accumulate with storage time and increase post-transfusion concentrations of inflammatory markers in the transfusion recipient [25]. This is likely through the direct interaction with monocytes, initiating a pro-inflammatory cascade, and subsequent interactions with T cells. Further, components of the complement cascade such as complement receptor 1 have also been identified in RBC microparticles [70]. Critically, microparticles result in the increased endothelial expression of ICAM-1 and E-Selectin and procoagulant activity [77]. However, no studies have specifically reported the influence of donor sex on extracellular vesicles and microparticles and recipient responses to transfusion exposure.

In summary, there is increasing data supporting biologically plausible pathways by which donor sex related differences in RBC characteristics, immunomodulatory and endothelial responses to transfusion exposure could result in adverse outcomes. However, further exploration of potential biological mechanisms should be an important consideration of future planned studies. 

## 5. Design Considerations for Future Clinical Studies in the Preterm Newborn

Blood donors, RBC preparation and transfusion practice has been given significant attention. However, more work is required to unravel the complex linkage between these critical three determinants of transfusion safety. These three factors, termed the “transfusion continuum” [50], require an integrated research approach. Such an approach, focusing on donor variation effects on RBC storage lesion and transfusion efficacy, has recently compared control and beta-thalassemia minor RBC donors [79]. Generating such a “triangular” strategy for studying the links comprising the transfusion chain, namely the donor, blood product, and the recipient, is worth consideration in the preterm newborn. 

As the available evidence supporting an effect of blood donor sex on outcomes in transfusion recipients is observational, there remains a compelling argument for further establishment and development of large ‘vein-to-vein’ data sets that would allow for robust evaluations of the clinical impact of donor characteristics. Linked donor/product and recipient data has been successfully used to explore the impact of donor sex on outcomes in adult transfusion recipients, yet this approach is under-utilized in the newborn. For example, even the most developed, The Scandinavian Donations and Transfusions (SCANDAT) database including over 21 million transfusion records and over 3.5 million donors, has yet to investigate donor effects in the preterm newborn [80]. 

While one logical evolution of such resources would be real time capture and reporting, another approach is the use of a novel analytical methodology to interrogate large-scale retrospective datasets. Brunn-Rasmussen and colleagues have recently employed causal inference methodology and targeted machine learning approaches to emulate the large scale randomized controlled trials investigating donor sex and transfusion recipient outcomes that are currently lacking [81]. The authors argue that “in the absence of high-powered randomized controlled trials (RCTs), explicitly emulating target trials using real-world data and targeted learning may provide better evidence on the effect of donor sex on patient mortality” [81]. There is also the opportunity to utilize data from recent large, multi-centre NICU transfusion studies such as the National Institute for Child Health and Human Development’s Transfusion of Prematures Trial or the Effects of Liberal vs. Restrictive Transfusion Thresholds on Survival and Neurocognitive Outcomes in Extremely Low-Birth-Weight Infants (ETTNO) trial for secondary analysis, including the role of donor sex [68,82]. 

The fact that each donor’s blood is biologically different, rendering the standardization of blood products at any level extremely difficult [50], has confounded clinical and laboratory-based studies examining transfusion adverse events. Further, observational data cannot be interpreted as evidence of a causal relationship due to the high risk of bias. As described earlier, to conclusively address this research question, three main sources of bias need to be controlled for: confounding by indication, co-interventions such as transfusion of other blood products and classification of exposure. Therefore, the only way to minimize bias, balance underlying prognostic factors related to the outcome of interest and have a homogenous intervention group is to conduct an RCT. Current clinical practice means the when the treating clinician orders an RBC transfusion for a patient, they do not know if it will come from a male or female donor. However, this quasi-randomization is inferior to double-blinded allocation within the construct of an RCT, where exposure can be carefully controlled. Currently, such an RCT is being conducted with adult patients (*n* = 8850) randomised to receive male-only or female-only donor transfusions, [83]. The primary objective of the innovative Trial Assessing Donor Sex on Recipient Mortality (iTADS) is to investigate the impact of a male-only transfusion strategy on survival compared to a strategy of female only donors in all hospital patients requiring at least one RBC transfusion. It is powered to detect an absolute risk reduction for death of 2% (30% to 28%); however, despite being large and well-designed, the trial will be underpowered if the effect is influenced by recipient sex. Furthermore, the limitation of uncontrollable stock levels of RBCs in the study blood banks has imposed a non-compliance provision in the protocol of 11%. Such a study design could however be easily transferred to the preterm neonatal population with the potential to investigate the impact of receiving blood from exclusively female or exclusively male donors and/or sex-matched transfusion.

Finally, RCTs are typically based on a biological hypothesis with the trial designed to prove or refute the hypothesis. As the mechanism/s by which donor sex affects clinical outcomes may be complex and multifactorial, we propose that simultaneous exploration of potential biological mechanisms within the context of a trial would represent an efficient approach to address significant knowledge gaps. Further, there is emerging evidence of the potential for additional blood product processing, such as RBC washing, to protect against transfusion related immune responses in the preterm newborn [84]. Confirmation of which mechanisms contribute to donor sex-related adverse outcomes could therefore also inform the design of future studies of alternative approaches in improving transfusion safety. This is particularly important, as confirmation of a donor sex effect would result in a significant change in the definition of RBC compatibility, and risk mitigation strategies would require operational changes for blood suppliers and hospitals to ensure adequate blood supply with minimum waste.

## 6. Conclusions

Within the newborn literature, the focus of transfusion research has predominantly been on factors influencing blood quality or the impact of strategies to control transfusion exposure. While there is ongoing controversy about the contribution of transfusion exposure to their adverse outcomes in the preterm newborn, donor characteristics are increasingly recognized as being potentially critical to transfusion recipient outcomes and are an emerging area of research. Systematic reviews have identified blood donor sex as one such factor that may influence transfusion recipient health, yet the limited evidence for such an effect in extremely preterm newborns remains conflicting. This is predominantly because all the available evidence is observational with inherent limitations. Furthermore, the biological mechanisms which could underlie the association of donor sex and increased risk of adverse outcomes following exposure to RBCs remain poorly characterized. 

Considering that preterm newborns remain a heavily transfused population, and that better donor selection may be one approach to significantly improving both morbidity and mortality in this high-risk group, further studies focusing on this specific question are warranted. Further larger cohort studies enabling the establishment and development of the large donor-recipient data sets would allow for further evaluation of the clinical impact of donor characteristics. However, the argument for well designed, adequately powered RCTs which allow for simultaneous exploration of potential biological mechanisms within the context of the trial is compelling. The knowledge generated from this type of research has the potential to change the way donors are selected and could improve patient outcomes. Such a practice change would entail significant operational changes for the blood supply but would represent a first step towards personalized transfusion therapy for the preterm newborn.

## Data Availability

Not applicable.

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
