# Peer review of "Red Blood Cell Donor Sex Associated Effects on Morbidity and Mortality in the Extremely Preterm Newborn"

_children, 2022, doi:10.3390/children9121980_

Round 1

Reviewer 1 Report

Dear Authors,

The results from different sources are compared regarding the donor sex and incidence of neonatal morbidities and mortality.

Red blood cell transfusions from female donors have been suggested in some literature reports to be associated with higher mortality rates.

In this regard, I have the following question: Whether the authors have encountered data in the literature on the influence of the effect of the four phases of the female menstrual cycle on RBC transfusion?

There are some typos and spelling errors in the manuscript that need to be corrected.

Author Response

We thank the reviewer for their interesting question. While there is limited data on the effect of sex-hormone consumption in pre-menopausal women (in the form of the oral contraceptive pill) - a reduction in storage related hemolysis but an enhanced susceptability to oxidative hemolysis- to our knowledge there is no data on the influence of the phase of the female menstrual cycle on RBC transfusion related outcomes.

The manuscript has been proof read and typographical errors corrected.

Reviewer 2 Report

In their manuscript entitled "The impact of red blood cell donor sex on recipient morbidity and mortality in the extremely low preterm newborn" Crawford et al., focus on the impact of donor's sex upon post-transfusion complications in preterm newborn recipients. This study is important for the field of blood transfusion and the universal effort to move towards a personalized blood transfusion. The study is well written but in this Reviewer's opinion it needs some alterations and additions before publication.

1. Introduction section: I think that the authors should also mention the link between storage lesion and post-transfusion outcomes in the paragraph that focuses on factors that influence post-transfusion outcomes. Here are some review studies they could mention: 10.2450/2017.0313-16 and 10.2450/2019.0217-18

2. Introduction section: I think, as they do in the Conclusion section, the authors should highlight in the final lines of the Introduction section the importance of moving towards a personalized transfusion therapy.

General comment: I think that as a reader I would like for a paragraph to be added to better understand the specific recipient background. This paragraph could describe the distinct physiology/biology of preterm newborns as recipients (e.g., their systematic immaturity, their hormone profile, their antioxidant defenses). I hereby suggest some references to aid the authors, but of course they can choose whatever fits them best from the bibliography: 10.3390/antiox10111672 , 10.1038/s41390-018-0003-2 and 10.1136/archdischild-2021-323433

3. Paragraph 4.1 The authors could add some more biologically and clinically significant information regarding differences between female and male donated RBCs. Recent studies have shown alterations in their redox , as well as in their membrane properties (refs: 10.2450/2020.0141-20 and 10.3324/haematol.2021.278895). Moreover, female stored red blood cells seem resistant to mechanical lysis post exposure to conditions that mimic the stress they will be implemented to in the circulation (10.1111/j.1423-0410.2010.01365.x). 

4. Paragraph 5: A similar integrated research approach was recently proposed as a triangular analysis (donor - blood bag - recipient), to simultaneously examine the three links of the transfusion chain and comprehend their biological and clinical impact upon each other. I think it is proper to mention this publication (10.3390/biomedicines10030530).

Overall, the authors should be praised for their great addition to the field of hematology/transfusion therapy.

Author Response

We thank the reviewer for this very constructive external review. All the points raised, and suggestions made, have been addressed and are outlined below. We feel that these changes to the manuscript have resulted in significant improvements.

1. Introduction section: I think that the authors should also mention the link between storage lesion and post-transfusion outcomes in the paragraph that focuses on factors that influence post-transfusion outcomes – We thank the reviewer for this suggestion. More focus has been placed on the role for storage lesion and the relevant paragraph amended (lines 40-50).

2. Introduction section: I think, as they do in the Conclusion section, the authors should highlight in the final lines of the Introduction section the importance of moving towards a personalized transfusion therapy – We thank the reviewer for this suggestion. The final sentences of the introductory section have been amended to make it clear to the reader that future studies, including those focusing on the biological mechanisms underlying any association between donor sex and transfusion recipient outcome, should be viewed as steps towards personalized transfusion therapy (lines 56-58)

General comment: I think that as a reader I would like for a paragraph to be added to better understand the specific recipient background. This paragraph could describe the distinct physiology/biology of preterm newborns as recipients (e.g., their systematic immaturity, their hormone profile, their antioxidant defenses). I hereby suggest some references to aid the authors, but of course they can choose whatever fits them best from the bibliography – We thank the reviewer for this suggestion and have added a paragraph to the introduction to directly address this (lines 40-65)

3. Paragraph 4.1 The authors could add some more biologically and clinically significant information regarding differences between female and male donated RBCs. Recent studies have shown alterations in their redox, as well as in their membrane properties. Moreover, female stored red blood cells seem resistant to mechanical lysis post exposure to conditions that mimic the stress they will be implemented to in the circulation – We thank the reviewer for this suggestion. The paragraph has been revised to discuss the inherent differences more completely in RBCs from male and female donors and now includes information on differences in ROX/antioxidant capacity and membrane properties (lines190-192, 196, 198-205)

4. Paragraph 5: A similar integrated research approach was recently proposed as a triangular analysis (donor - blood bag - recipient), to simultaneously examine the three links of the transfusion chain and comprehend their biological and clinical impact upon each other. I think it is proper to mention this publication (10.3390/biomedicines10030530) – We thank the reviewer for highlighting this recent publication. The relevant paragraph has been revised to include this reference and highlight the potential for this approach to provide valuable information within the preterm newborn population (lines283-288).

Round 2

Reviewer 2 Report

I would like to thank the authors for the extensive reconstruction of their manuscript. All my concerns have been exceptionally addressed. I strongly believe that this manuscript will be of great value to the scientific community.